# Effects of rebound exercises on balance and mobility of people with neurological disorders: A systematic review

**Adaora Justina Okemuo**[1,2]\*, **Dearbhla Gallagher**[3], **Yetunde Marion Dairo**[1]

**1** School of Health and Social Care Professions, Buckinghamshire New University, High Wycombe, United Kingdom, **2** Department of Medical Rehabilitation, University of Nigeria, Enugu Campus, Nsukka, Nigeria, **3** School of Human and Social Sciences, Buckinghamshire New University, High Wycombe, United Kingdom

\* Adaora.okemuo@bucks.ac.uk

**Data Availability Statement:** All relevant data are available on OSF: https://doi.org/10.17605/OSF.IO/C75G3.

**Funding:** The authors received no specific funding for this work.

## Abstract

### Background

Therapeutic rebound exercise is gaining popularity among the general population, but its effectiveness in individuals with neurological impairments remains uncertain. To shed light on this, a systematic review was conducted between November 2021 and March 2023 to study the impact of rebound exercise on balance and mobility in this group.

### Methods

Six databases were searched. Studies were included if written in English, peer-reviewed, had original research data and assessed the effect of rebound exercise in adults with neurological disorders. The outcomes measured were balance and mobility. Two reviewers independently appraised study quality using the Critical Appraisal Skills Program for Randomized Controlled Trials. Finally, a meta-summary of the included studies was completed, and a meta-analysis was performed using RevMan software version 5.3 to determine the effectiveness of the intervention.

### Results

Five studies were included comprising 130 participants aged 31.32±7.67 to 58±12 years, 72% male and 28% female. Participants were in-patients with stroke (49%), multiple sclerosis (24%), Parkinson's disease (15%) and spinal cord injury (12%). The included papers had moderate to high methodological quality. The timed up-and-go test revealed that the rebound group participants could walk 6.08 seconds quicker over three to eight weeks. Pooled results show that rebound exercise significantly improves mobility (-0.53[-0.94, -0.11], p = 0.01), but no significant improvement was observed in balance.

### Conclusion

Rebound exercise has shown the potential to improve mobility in people with neurological disorders. However, the findings should be in the context that the included

**Competing interests:** The authors have declared that no competing interests exist.

studies are few and participants were in in-patient settings. PROSPERO registration: CRD42021298030.

## Introduction

Individuals with neurological disorders experience various challenges, such as movement impairments, balance issues, fear of falls, reduced exercise tolerance, loss of muscle strength, reduced functional independence, and lower quality of life [1]. These impairments, particularly movement and balance dysfunction, can hinder their overall well-being and make physical activity difficult [2, 3]. World Health Organisation recommends that adults engage in regular physical activity to maintain a healthy life and reduce the risk of chronic diseases such as diabetes mellitus, obesity, cardiovascular disease, cancers, and stroke [4, 5]. This recommendation is relevant for all adults, including those with neurological disabilities [6, 7]. However, the physical impairments and mobility limitations associated with neurological disorders limit physical activities [8], thus making them less physically fit. Therefore, finding ways to improve movement in this population is crucial. Rebound exercise is one such approach that has shown promise in the general population [9–17], but its effectiveness in the neurological population requires further investigation, which is the aim of this review.

Bouncing on a mini trampoline, also known as rebound exercise, is gaining popularity in the healthcare industry as a promising therapeutic modality. This exercise involves repetitive vertical movement of the body on a compliant surface, offering flexibility, adaptability, safety, and gratification [9, 10]. Unlike other forms of exercise, rebound exercise uses gravity to optimise gains while conserving efforts, making it the most potent form of cellular exercise [11, 12]. The National Aeronautics and Space Administration (NASA) found rebound exercise to be 68% more efficient than running as it expends less energy and places less stress on the cardiovascular system and joints [11]. Anecdotal evidence suggests that rebound exercise is enjoyable and sustainable, promoting adherence to exercise. Moreover, individuals can enjoy this form of exercise without experiencing the intensity of a typical workout [9]. Studies have examined the impact of rebound exercise on various groups of people, analysing parameters like muscular strength and endurance [13, 14], balance and coordination [13, 15], athletic performance [16], BMI, and overall quality of life [9, 17]. The positive attributes of rebound exercise, such as being low impact [18] and time, cost, and energy efficient [11, 19, 20], have made it a successful option for certain populations, including those with intellectual disabilities [19], diabetes mellitus [21], and individuals who are overweight or obese [9, 17]. These findings provide evidence for the effectiveness of rebound exercise in these cohorts.

Despite the growing interest in using rebound exercise interventions for patient rehabilitation, a comprehensive synthesis regarding its specific effects in individuals with neurological disorders remains lacking. This knowledge gap impedes our understanding of rebound exercise's potential benefits, safety, and feasibility as a therapeutic modality for this population. While a limited number of studies [22–26] have shown promising results of rebound exercise in individuals with neurological disorders, the absence of systematic reviews assessing its efficacy for neurorehabilitation is notable. Therefore, there is an urgent need for a comprehensive review that critically examines and synthesises the available evidence on the effects of rebound exercise in individuals with neurological disorders. By addressing this literature gap, the systematic review aims to provide clinicians, researchers, and policymakers with evidence-based insights to inform decision-making, enhance rehabilitation practices, and identify key areas for further research in this crucial field.

## Review question

What is the effect of rebound exercise on balance and mobility in people with neurological disorders?

## Methods

### Study design

This systematic review was registered with the International Prospective Register of Systematic Reviews (PROSPERO, CRD42021298030) and reported following the guidelines according to the Preferred Reporting Items for Systematic Reviews and Meta-Analysis (PRISMA) statement. No protocol for the systematic review was published.

### Inclusion/exclusion criteria

**Participants.**   Studies with adults aged 18 years and above diagnosed with a neurological disorder were included. Studies on children and adolescents (under 18 years of age) and non-human participants were excluded.

**Intervention.**   Published studies focused on rebound exercise's effects, efficacy or effectiveness were included.

**Comparison/control.**   The control included people with neurological disorders receiving standard physiotherapy care. In contrast, the comparator group included other types of intervention. The control/comparison group differed from the intervention group for this review.

**Outcomes.**   The outcomes assessed were balance and mobility. There were no restrictions on the outcome measures used to assess these outcomes, provided they were standard objective, valid and reliable instruments.

**Type of studies.**   Studies that investigated the effect of rebound exercise were eligible, provided they were published in English. There were no restrictions based on the year of publication.

**Context.**   Studies conducted in hospital and community settings were eligible. There was no restriction based on geographical location.

### Search strategy

A thorough search was completed on several databases to identify all relevant papers on rebound exercise training for improving balance and mobility in people with neurological disorders. Medical subject headings (MeSH terms), keywords and phrases such as 'rebound exercise', 'rebound therapy', 'mini-trampoline exercise', 'trampoline', 'trampolining', 'neurological disorder', 'neurological disease', and 'neurological disability' helped to identify relevant studies. PubMed, SportDiscus, PsycINFO, ProQuest and Cochrane Library Trials were searched in combination with database-specific filters for controlled trials, where applicable. Searches for relevant studies were also conducted on grey literature sources like Google Scholar, clinical-trials.gov and open thesis. Author experts in this field were contacted via email to enquire about relevant unpublished studies. To capture substitute terms, we used truncation (*) and Boolean operators (OR/AND) to connect terms within and between concepts. Additionally, we performed a manual search of journals and citation chaining of the reference lists of the already identified studies. We conducted the first search in November 2021 and re-ran the search in July 2022 and March 2023 to ensure we missed no newly published studies. The search identified publications from inception to March 2023. An example of a detailed search strategy is shown in S1 Appendix.

## Data extraction

Results of the literature search were exported into Mendeley to check for duplication of studies. The screening was conducted independently on the title and abstract of articles retrieved from the databases and manual reference searches by two review authors. Potentially eligible studies underwent further screening independently by two reviewers (AO & YD). This screening was done by reading through the full texts of the selected articles using the eligibility criteria. Any disagreement between the reviewers over the eligibility of studies was settled either through deliberations or consulting with a third reviewer (DG). Standardised data extraction forms were pre-designed and pre-piloted by the authors on two unrelated studies before being used for the primary data extraction and evidence synthesis. Extracted information included: the study setting, population, participant demographics, details of the intervention and control conditions, outcome measures, key findings, and evaluation of the risk of bias. Two review authors extracted data independently; discrepancies were identified and resolved through discussion. We requested missing data from the study authors through email correspondence. The selection process details are presented in a PRISMA flowchart, including the reasons for excluding studies (Fig 1).

## Risk of bias assessment

The Critical Appraisal Skills Programme (CASP) Randomized Controlled Trial checklist [27] was used independently by two review researchers to assess the risk of bias in the individual studies included in the systematic review. It consists of eleven questions broadly divided into four sections appraising various aspects of the study, including randomisation integrity, methodological soundness, the correctness of the result and the significance of the study to the locality. Each question of the CASP tool has three responses ranging from 'yes', 'no' and 'cannot tell'. The total number of 'yes' responses was used to calculate the percentage risk of bias. Scores under 30% represent a high risk of bias/low-quality study, 30–70% represent a moderate risk of bias and above 70% represent a low risk of bias/high-quality study [28].

## Data analysis

The Review Manager (RevMan) statistical software by the Cochrane Collaboration (version 5.3) was used for the data synthesis of this systematic review. Effect sizes of mean difference and standardised mean difference were calculated by RevMan and represented on forest plots. Chi-squared test of homogeneity was conducted to generate the inconsistency index ($I^2$) statistics on the mean of assessed outcomes of included studies. Meta-analysis was performed if the studies had acceptable heterogeneity ($I^2 < 50\%$). We assumed heterogeneity in actual effect sizes and thus used the random effects model with a 95% confidence interval to pool the effect sizes.

# Results

A total of 858 studies were identified from searching six databases. Sixty-eight (68) duplicates were removed, and 729 and 58 studies were excluded after abstract and full-text screenings, respectively (see Fig 1). We included five studies published between 2013 and 2019 in the systematic analysis, and only three were eligible to be pooled in a meta-analysis. The $I^2$ test revealed considerable heterogeneity among four studies ($I^2 > 50\%$), which was instrumental in selecting the combinable studies for meta-analysis (S2 Appendix). Two studies [22, 23] were excluded as they did not meet the inclusion criteria for meta-analysis, which required trials assessing the same outcomes and comparable outcome measures.

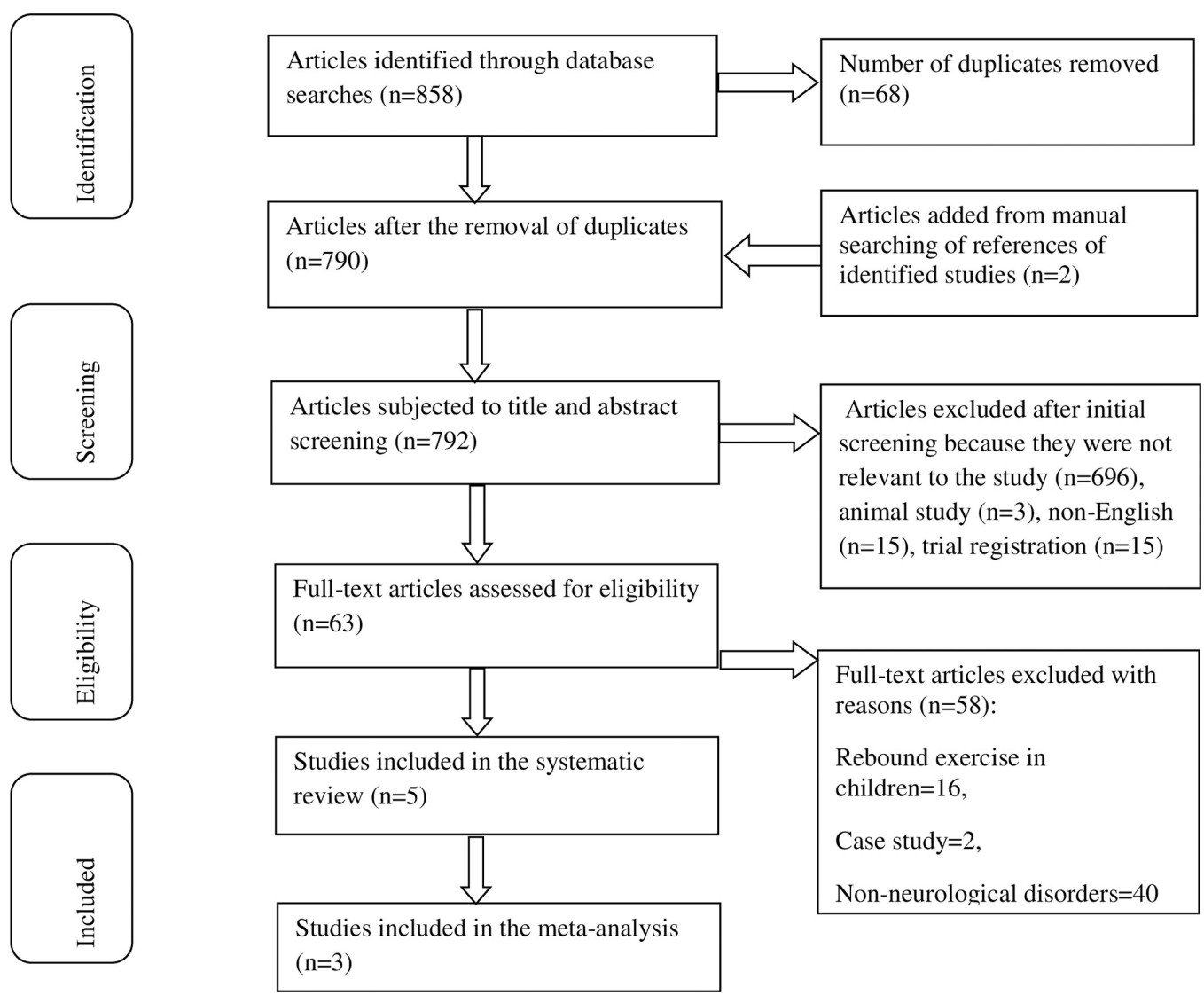

**Fig 1. This is the PRISMA flowchart.**

## Risk of bias in included studies

Quality assessments of the studies are summarised in Table 1, while the details of the studies and participants are presented in Tables 2 and 3. The methodological quality of the studies ranged between moderate (55–64%) [23, 25, 26] and high quality (82–91%) scores [22, 24] on the CASP checklist. There was evidence of reporting bias as only one study reported adequate information to ascertain if appropriate randomisation and allocation concealment were conducted [24]. Furthermore, blinding of the participants, investigators and outcome assessors to the intervention given was considered as reported in just two studies [22, 24]. All studies except for one [26] accounted for all participants who entered the study on conclusion.

## Meta summary of included studies

A total number of one hundred and thirty (130) participants took part in the five included studies, with mean ages ranging from 31.32±7.67 to 58±12 years (Table 2). The total

**Table 1. A quality appraisal of studies.**

| S/N | Questions | Miklitsch et al, 2013 | Sisi et al., 2013 | Hahn et al., 2015 | Daneshvar et al., 2019 | Sadeghi et al., 2019 |
|---|---|---|---|---|---|---|
| 1 | Did the study address a clearly focused research question? (PICO addressed?) | Yes | Yes | Yes | Yes | Yes |
| 2 | Was the assignment of participants to interventions randomised? (Was randomisation done appropriately and concealed from the participants and investigators?) | Yes | Cannot tell | Cannot tell | Cannot tell | Cannot tell |
| 3 | Were all participants who entered the study accounted for at its conclusion? (Loss to follow-up, intent-to-treat analysis?) | Yes | Yes | No | Yes | Yes |
| 4 | Were the participants, investigators and outcome assessors blinded to the intervention given? | Yes | Cannot tell | Cannot tell | Yes | Cannot tell |
| 5 | Were the study groups similar at the start of the randomised controlled trial? | Yes | Yes | Yes | Yes | Yes |
| 6 | Apart from the experimental intervention, did each study group receive the same level of care (that is, were they treated equally)? | Yes | Yes | Yes | Yes | Yes |
| 7 | Were the effects of the intervention reported comprehensively? (power calculation, clear outcomes, p-values reported) | Yes | Yes | No | Yes | Yes |
| 8 | Was the precision of the estimate of the intervention or treatment effect reported? (Confidence intervals reported) | No | No | No | No | No |
| 9 | Do the benefits of the experimental intervention outweigh the harms and costs? (Harms or unintended effect of intervention reported) | Yes | Cannot tell | Yes | Yes | Cannot tell |
| 10 | Can the results be applied to your local population/ in your context? (Was the right population used, and proper outcomes assessed?) | Yes | Yes | Yes | Yes | Yes |
| 11 | Would the experimental intervention provide greater value to the people in your care than any existing interventions? | Yes | Yes | Yes | Yes | Yes |
| | Total number of YES responses | 10 (91%) | 7 (64%) | 6 (55%) | 9 (82%) | 7 (64%) |

participants in the studies were primarily men (72%). Nearly all the included studies were randomised controlled trials [23–26], with only one quasi-experimental study [22]. Four studies were two-arm trials, while one study [25] was a three-arm trial involving rebound vs Pilates vs control. Within the two-arm trials, two studies had the groups; rebound intervention vs control [23, 26], while the remaining two had a comparative group in place of control, rebound vs balance training [24] and rebound vs treadmill [22]. Trained physiotherapists delivered the interventions in all the studies. Regarding the disease type, two studies (49%) involved stroke survivors [24, 26], and the rest involved people with multiple sclerosis (24%) [25], Parkinson's disease (15%) [22] and spinal cord injury (12%) [23]. The duration of the intervention varied considerably across studies ranging from three weeks to twelve weeks. However, the duration and frequency of the intervention were mainly uniform, with an average of 30 minutes per session, three times weekly. Three studies [24–26] assessed balance and mobility using the Berg balance scale and timed up-and-go (TUG) test, while a fourth study [23] assessed balance using the Kistler force plate. The fifth study [22] assessed mobility through the knee joint

**Table 2. Characteristics of included studies.**

| Study (Country) | Disease | Sample size (I/C) | Mean age (x±sd) | Sex (% m/f) | Setting | Duration of intervention |
|---|---|---|---|---|---|---|
| Miklitsch et al, 2013 (Germany) | Stroke | 20/20 | 58.0±12 | 62.5/ 37.5% | Hospital | 3 weeks |
| Sisi et al, 2013 (Iran) | Multiple sclerosis | 15/15 | 31.32±7.67 | 100/ 0% | Not reported | 8 weeks |
| Hahn et al., 2015 (Korea) | Stroke | 12/12 | 54.48±10 | 58.3/ 41.7% | Hospital | 6 weeks |
| Daneshvar et al., 2019 (Iran) | Parkinson's disease | 10/10 | 56.4±7.45 | Not reported | Hospital | 8 weeks |
| Sadeghi et al., 2019 (Iran) | Spinal cord injury | 8/8 | 36.15±6.05 | 68.75/ 31.25% | Rehabilitation centre | 12 weeks |

I/C- Intervention/control, x±sd- mean±standard deviation, m/f- male/female

**Table 3. Main findings.**

| Study ID | Outcome measures | Intervention | Control | Key findings |
|---|---|---|---|---|
| Miklitsch et al, 2013[r] | Mobility-TUG, Balance-BBS, Activities of daily living- Barthel index | Rebound exercise: 10 sessions with 30 minutes per session for 3 weeks. TUG-14.2 ±8 (P = 0.347)*, BBS-52.66±3.97 (P = 0.004)*, ADL- 83.66±23.81(P = 0.157) * | Balance training: 10 sessions with 30 minutes per session for 3 weeks. TUG- 21.5±11, BBS- 47±7.94, ADL- 72.66±25.4 | Significant increase in balance and increase (but not statistically significant) in the mobility and ability to perform activities of daily living in the rebound group compared to the control |
| Sisi et al., 2013[r] | Dynamic balance/ mobility-TUG, Static balance-BBS | G1: Rebound exercise: 24 sessions with 30 minutes per session for 8 weeks. TUG-11.43±2.37, BBS- 36.06±2.12 G2: Pilates exercises: 24 sessions with 30 minutes per session for 8 weeks. TUG-11.72±3.01, BBS-38.43±2.87 | Routine exercises TUG-12.23±1.81 (P = 0.01)* BBS-31.0±2.49 (P = 0.01)* | Both rebound and Pilate exercises significantly improved balance compared to the control, but the rebound was more effective in improving dynamic balance, while Pilate was more effective in improving static balance. |
| Hahn et al., 2015[r] | Dynamic balance- TUG, Static balance- BBS | Rebound exercise: 3 times a week with 30 minutes duration per session for 6 weeks + routine physiotherapy TUG-26.3±11, BBS-44.3±7.5 (P<0.05)* | Routine physiotherapy: 3 times weekly with 30 minutes per session for 6 weeks. TUG-30.7±11.2, BBS-41.8±6.3 | Compared to the control, balance, dynamic gait and falls efficacy significantly improved in the rebound group. |
| Daneshvar et al., 2019[q] | Quality of life- PDQ-39 | Rebound exercise: 3 sessions a week with 20–45 minutes per session for 8 weeks 147.6±13.22 (P<0.001)* | Treadmill training with body weight support: 3 sessions a week with 20–45 minutes per session for 8 weeks. 118.38±12.48 | Although both exercises improved function in participants, rebound exercises showed more significant improvement in range of motion, proprioception and quality of life compared to treadmill exercises. |
| Sadeghi et al., 2019[r] | Kistler force plate for COP-based parameters | Rebound exercise: 3 sessions a week with 10–30 minutes per session for 12 weeks | Traditional exercises | The rebound group significantly improved static balance/ stability compared to the control group. |

r- randomized controlled trial, q-quasi-experimental, values presented as mean±standard deviation, ()*-significance level between rebound and control groups.

range of motion using a goniometer. In addition to this, they also measured the participants' proprioception and quality of life, outcomes, which none of the other four measured. The outcomes were assessed at only two points during the study (start and end of the trial) across all five studies. Of the five studies, three were hospital-based [22, 24, 26], one was community-based [23], and the last [25] did not report the context. However, the studies pooled in the meta-analysis took place in the hospital.

## Effect of rebound exercise on balance

Of the four studies that reported on balance, one study [23] used a different outcome measure, force plate, that was not comparable with the rest (Chi$^2$ = 27.88, I$^2$ = 89% (p<0.000001)) and thus was not part of the meta-analysis. However, the excluded study reported that participants in the rebound group had significantly more improved balance than the control (20.6±8.4 vs 100.13±35.9; p = 0.05).

The balance scores for the three remaining studies were homogenous in a homogeneity test (Z = 6.63; Chi$^2$ = 0.90; I$^2$ = 0%), so a meta-analysis was done. Data pooled from the three studies favoured the control group with a mean effect size of 4.97 (95% CI 3.50–6.43; p< 1x $10^{-5}$) (Fig 2).

## Effect of rebound exercise on mobility

Three studies assessed mobility and were comparable on homogeneity analysis to be pooled in a meta-analysis (Chi$^2$ = 0.74; I$^2$ = 0%). TUG results from the studies decreased from a mean baseline time of 23.39±9.06 seconds to 17.31±7.12 seconds post-test in the rebound group and a decrease from 24.64±9.35 seconds to 21.48±8 seconds in the control group. The participants in the rebound group walked quicker by an average of 6.08±6.08s over three to eight weeks,

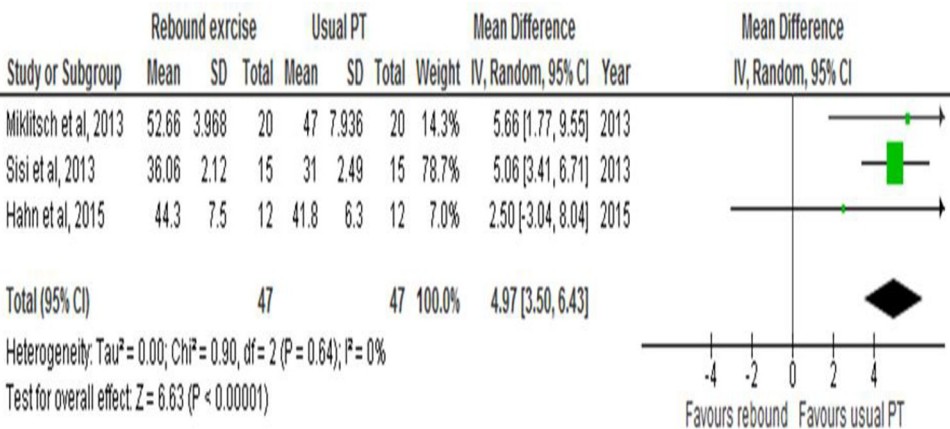

**Fig 2. This is the forest plot for the effect of rebound exercise on balance.**

with a study [24] recording as much as a 10.2±8 seconds decrease in walking time. The forest plot in Fig 3 shows that the rebound group has an edge over the control group, indicating a statistically significant improvement in walking time for the rebound group with a pooled effect of -0.53 (95%CI -0.94, -0.11; p = 0.01).

## Discussion

In this review, five studies were examined to determine the effect of rebound exercise on the balance and mobility of individuals with neurological disorders. However, only a few studies met the meta-analysis inclusion criteria. Despite the small number of studies, the meta-analysis displayed significant improvement in mobility due to rebound exercise. All studies included in the meta-analysis showed that participants' mobility improved with this exercise. Results from the TUG test revealed that the rebound exercise group's test time decreased by almost twice as much (6.08s) compared to that of the control group (3.18s), in a ratio of 1.92:1. The decrease in TUG test times indicates an enhanced ability to stand up from a chair more quickly, walk faster with better balance, turn better, and sit down with greater control without assistance.

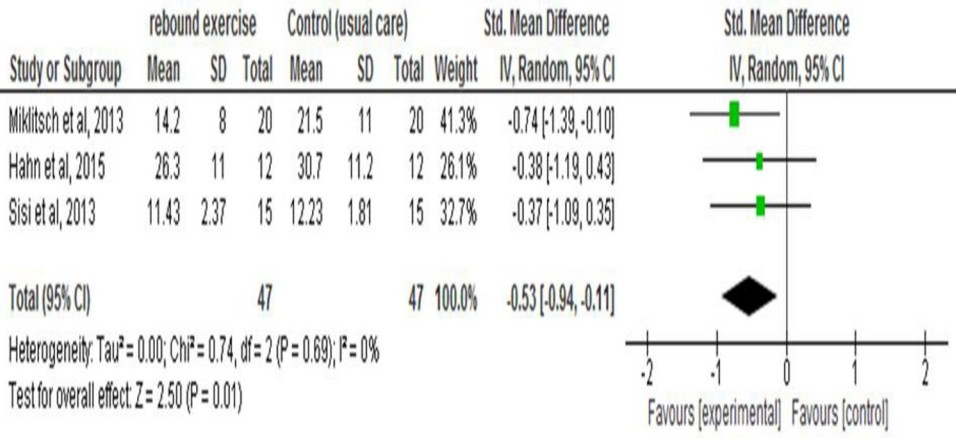

**Fig 3. This is the forest plot for the effect of rebound exercise on mobility.**

The improvement may stem from one or a combination of these aspects, but the studies did not record them specifically.

The improved mobility was observed between three to eight weeks. Two included studies, Hahn et al. [26] and Sisi et al. [25], reported a significant increase in mobility in the rebound group after six and eight weeks of intervention, respectively. In contrast, the third study by Miklitsch et al. [24] showed a significant improvement in mobility for both the rebound and control groups after three weeks of exercise but with no significant difference between the two groups. It is likely that the relatively short duration of the trial did not allow for the full effect of rebound exercise to become evident. Other studies have also reported an increase in mobility with aerobic exercise. For instance, a systematic review of nine randomised controlled trials showed that aerobic exercise like walking, treadmill, and stationary cycling significantly improved mobility among stroke survivors [29]. Similarly, Koop et al. [30] demonstrated an increase in functional mobility among Parkinson's disease patients after eight weeks of high-intensity aerobic exercise on a stationary recumbent bicycle at a 30% augmented cadence. Bouncing on a trampoline engages major muscle groups in the upper limbs, trunk, and lower limbs, stimulating and strengthening these muscles through constant vertical movement against the gravitational force [31]. Furthermore, maintaining stability on the soft, unstable surface of the trampoline requires complex sensorimotor stimulation necessary for balance and muscle strength [26, 32].

Surprisingly, rebound exercise did not significantly improve the participants' balance, unlike mobility. This contradicts previous studies that show that rebound exercise can enhance balance and stability for people of all ages. For instance, research conducted by Márquez et al. [33] and Atilgan [34] demonstrates that trampoline significantly improves balance and stability for athletes and young boys. Similarly, rebound exercise has been found to increase balance and mobility in older adults [13]. The individual studies [24–26] included in this review also reported a significant increase in balance for the rebound group compared to the control, which is consistent with the wider population's results. One likely explanation for this difference is the relatively small size of the individual studies, which may have resulted in a type 1 error in reporting a significant increase in balance where there is none [35]. Additionally, the balance scale used in these studies, the Berg balance scale, may not be sensitive enough to detect important changes despite its acceptable validity and reliability [36]. The scale's varying reliability and ceiling effect could also contribute to this limitation [36]. Another factor to consider is that all the studies were conducted on in-patients, limiting opportunities to challenge their balance.

This review revealed that the current literature on rebound exercise and its impact on neurological disorders has a notable limitation. It primarily relies on experimental trials and lacks observational or cohort studies examining rebound exercise's broader effects. The existing research has primarily focused on measuring intervention effectiveness, neglecting other pivotal factors such as acceptability, adherence, and participants' perceptions [37]. It is crucial to recognise that low acceptability or adherence can significantly compromise an intervention's overall impact, regardless of its effectiveness [38]. Therefore, there is an urgent need to conduct observational and qualitative studies to comprehensively explore the holistic potential of rebound exercise for individuals with neurological disorders.

This review further highlights the lack of information on the practicality of rebound exercise for individuals with neurological conditions in a community setting. The studies reviewed were conducted in hospitals, with participants engaging in rebound exercises three times a week. However, it is unclear if this frequency can be maintained outside of a hospital environment. Additionally, there is a need for further research to determine the efficacy of rebound exercise on other functional aspects. To fully assess the feasibility of rebound exercise for

community-dwelling adults with neurological disorders, studies should also consider patients' personal experiences, perceived effectiveness, and willingness to participate in the regimen. These factors will influence their motivation and adherence to the exercise program.

## Limitations

Despite being the first systematic review to investigate the available literature on the efficacy of rebound exercise in people with neurological disorders, the scope of evidence is limited as the included studies are few. Another limitation is that the studies included in the review had varying intervention durations and outcome measures. However, the impact of this limitation on the findings was minimised by using the heterogeneity test to combine only homogenous studies. A further review limitation is the lack of specificity of the neurological disorders studied. This is noteworthy because the clinical presentations and disease course of various neurological disorders differ, which may influence the outcomes. The differing trajectories of these disorders can impact the outcomes and make comparisons challenging, as individuals may be at different stages of their disease. Future reviews should focus on specific types of neurological disorders to provide more definitive results, considering the unique characteristics and needs of individuals with different disorders. Finally, publication bias may have influenced the results, as most included studies reported positive effects of rebound exercise, and there needed to be more studies to calculate asymmetry with the funnel plot test.

## Implications for research and practice

The available evidence supports the use of rebound exercise in improving the mobility of neurological patients within the hospital. Improvement in walking time is clinically important as an objective measure of treatment progress and indicates a reduced risk of falls and increased independence. Further robust trials are needed to determine its feasibility and effectiveness in contexts other than the hospital before recommendations can be made for its use in community neurorehabilitation. Future studies should also consider looking into the participants' views, experiences, and acceptability of rebound exercises.

## Conclusion

Rebound exercise has shown the potential to improve mobility in people with neurological disorders. The review suggests that rebound exercise could improve walking time in adults with neurological disorders within the hospital setting. Still, it does not appear to improve balance in this group of people when compared to standard physiotherapy intervention. Whilst there were only a few trials in this review, it has demonstrated rebound exercise's effectiveness and potential as an adjunct in neurorehabilitation. However, it is unclear if rebound exercise would have the same effect in the community setting. Therefore, studies on the feasibility of rebound exercises in the community setting are required. Additionally, there is a need for studies investigating the effectiveness of rebound exercise on other aspects of function, particularly those relating to physical activities, as they may impact people's quality of life and independence.

## Supporting information

**S1 Appendix. This is the search strategy.**
(TIF)

**S2 Appendix. This is the Chi-squared test of heterogeneity.**
(TIF)

**S3 Appendix. This is the PRISMA checklist for systematic reviews.**
(PDF)

## Author Contributions

**Conceptualization:** Adaora Justina Okemuo.

**Data curation:** Adaora Justina Okemuo, Yetunde Marion Dairo.

**Formal analysis:** Adaora Justina Okemuo, Yetunde Marion Dairo.

**Investigation:** Adaora Justina Okemuo.

**Methodology:** Adaora Justina Okemuo, Dearbhla Gallagher, Yetunde Marion Dairo.

**Project administration:** Adaora Justina Okemuo.

**Resources:** Adaora Justina Okemuo, Yetunde Marion Dairo.

**Software:** Adaora Justina Okemuo.

**Supervision:** Dearbhla Gallagher, Yetunde Marion Dairo.

**Validation:** Dearbhla Gallagher, Yetunde Marion Dairo.

**Visualization:** Dearbhla Gallagher, Yetunde Marion Dairo.

**Writing – original draft:** Adaora Justina Okemuo.

**Writing – review & editing:** Adaora Justina Okemuo, Dearbhla Gallagher, Yetunde Marion Dairo.

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
