## [Decision Letter · Decision Letter 0]

19 Sep 2023

EFFECTS OF REBOUND EXERCISES ON BALANCE AND MOBILITY OF PEOPLE WITH NEUROLOGICAL DISORDERS: A SYSTEMATIC REVIEW

PONE-D-23-19518

Dear Dr. Okemuo,

We’re pleased to inform you that your manuscript has been judged scientifically suitable for publication and will be formally accepted for publication once it meets all outstanding technical requirements.

Kind regards,

Ragab Kamal Elnaggar

Academic Editor

PLOS ONE

Additional Editor Comments (optional):

Reviewers' comments:

Reviewer's Responses to Questions

**Comments to the Author**

1. Is the manuscript technically sound, and do the data support the conclusions?

Reviewer #1: Yes

2. Has the statistical analysis been performed appropriately and rigorously? 

Reviewer #1: Yes

3. Have the authors made all data underlying the findings in their manuscript fully available?

Reviewer #1: Yes

4. Is the manuscript presented in an intelligible fashion and written in standard English?

Reviewer #1: Yes

5. Review Comments to the Author

Reviewer #1: Dear author,

Manuscript is nicely written with Specifically serving the scientific aim. I think if possible then in future you should also work in this filed to prove the effectiveness of rebound exercises in specific neurological condition.

6. PLOS authors have the option to publish the peer review history of their article (what does this mean?). If published, this will include your full peer review and any attached files.

Reviewer #1: No
